# Are Vision Transformers More Data Hungry Than Newborn Visual Systems?

**Lalit Pandey**
Department of Informatics
Indiana University Bloomington
lpandey@iu.edu

**Samantha M. W. Wood**
Department of Informatics
Indiana University Bloomington
sw113@iu.edu

**Justin N. Wood**
Department of Informatics
Indiana University Bloomington
woodjn@iu.edu

## Abstract

Vision transformers (ViTs) are top-performing models on many computer vision benchmarks and can accurately predict human behavior on object recognition tasks. However, researchers question the value of using ViTs as models of biological learning because ViTs are thought to be more "data hungry" than brains, with ViTs requiring more training data to reach similar levels of performance. To test this assumption, we directly compared the learning abilities of ViTs and animals, by performing parallel controlled-rearing experiments on ViTs and newborn chicks. We first raised chicks in impoverished visual environments containing a single object, then simulated the training data available in those environments by building virtual animal chambers in a video game engine. We recorded the first-person images acquired by agents moving through the virtual chambers and used those images to train self-supervised ViTs that leverage time as a teaching signal, akin to biological visual systems. When ViTs were trained "through the eyes" of newborn chicks, the ViTs solved the same view-invariant object recognition tasks as the chicks. Thus, ViTs were not more data hungry than newborn visual systems: both learned view-invariant object representations in impoverished visual environments. The flexible and generic attention-based learning mechanism in ViTs—combined with the embodied data streams available to newborn animals—appears sufficient to drive the development of animal-like object recognition.

## 1 Introduction

Vision transformers (ViTs) have emerged as leading models in computer vision, achieving state-of-the-art performance across many tasks, including object recognition [4], scene recognition [5], object segmentation [9], action recognition [70], and visual navigation [14]. Recent studies also suggest that transformers share deep computational similarities with human and animal brains [8, 19, 45, 55]. For instance, transformers are closely related to current hippocampus models in computational neuroscience and can reproduce the precisely tuned spatial representations of the hippocampal formation (e.g., place and grid cells; [55]). ViTs also outperform convolutional neural networks (CNNs) on challenging image classification tasks and are more likely to learn human-like shape biases (and make human-like errors) than CNNs [18, 50].

Despite these strengths, there is a lingering worry about using ViTs as models of biological vision. The worry relates to the large amount of data needed to train ViTs, compared to the relatively small

37th Conference on Neural Information Processing Systems (NeurIPS 2023).

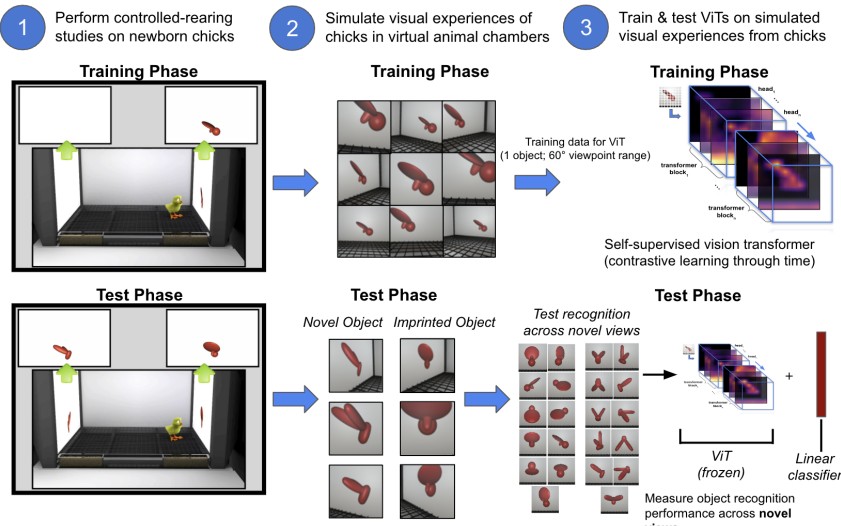

Figure 1: Our design had three steps: (1) Run controlled-rearing experiments testing how view-invariant object recognition develops in newborn chicks. (2) Simulate the visual experiences (training data) available to the chicks during the training and test phases, using virtual animal chambers constructed in a video game engine (Unity 3D). (3) Train and test the ViTs on the simulated visual experiences from the chick experiments. For training, we used self-supervised ViTs that leverage time as a teaching signal. For testing, we froze the ViTs and attached a linear classifier to the penultimate layer. We then trained & tested the linear classifier with the simulated test images from the chick experiments, in a cross-validated design.

amount of training data needed by biological brains. The ViTs that have revolutionized computer vision are trained on massive amounts of data (e.g., ImageNet, which contains millions of images across over 20,000 object categories). Conversely, animals do not require massive amounts of object experience to solve object perception tasks. For example, even when newborn chicks are reared in impoverished environments containing only *a single object* (i.e., sparse training data from birth), they still rapidly learn many object perception abilities, including segmenting objects from backgrounds [64], binding colors and shapes into integrated object representations [58], recognizing objects and faces from novel views [57, 66, 65, 68], and remembering objects that have disappeared from view [42]. Thus, ViTs appear to be more "data hungry" than newborn visual systems.

Here, we suggest that the learning gap between animals and ViTs is not as large as it appears. To compare learning across animals and ViTs, we performed parallel controlled-rearing experiments on newborn chicks and ViTs (Figure 1). First, we raised newborn chicks in strictly controlled virtual environments and measured the chicks' view-invariant object recognition performance (data reported in [57]). Second, we created digital twins (virtual simulations) of the controlled-rearing chambers in a video game engine, then simulated the chicks' visual training data by recording the first-person images acquired by an agent moving through the virtual chambers. Third, we trained ViTs on the simulated data streams from the virtual chambers and tested the ViTs with the same stimuli used to test the chicks. Consequently, the chicks and ViTs were trained on the same data and tested on the same task, allowing for direct comparison of their learning abilities.

## 1.1 Related Work

Our study uses a "digital twin" approach for comparing the learning abilities of newborn animals and machines [29, 30]. Digital twin experiments involve first selecting a target animal study, then creating digital twins (virtual environments) of the animal environments in a video game engine. We then train and test artificial agents in those virtual environments and compare the agents' behavior with the real animals' behavior in the target study. By raising and testing animals and machines in the same environments, we can determine whether they spontaneously learn common abilities when trained on the same data distributions. Digital twin experiments thus resemble prior studies that trained AI algorithms on egocentric (first-person) images from human babies and children, acquired from head-

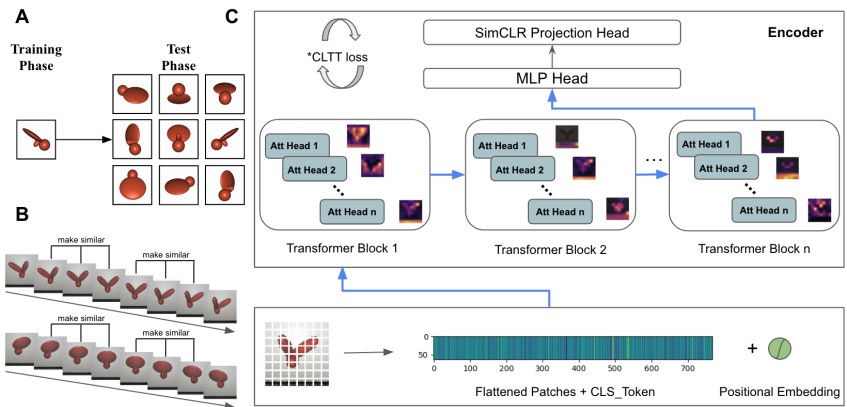

Figure 2: (A) View-invariant object recognition task. Newborn chicks and ViTs were trained in an impoverished visual environment (containing one object seen from a single viewpoint range), then tested on their ability to recognize that object across novel views. (B) ViT-CoT leverages the temporal structure of embodied data streams. Learning occurs by making successive views (images seen in a 300 ms temporal window) more similar in the embedding space. (C) The ViT architecture. The image is first divided into smaller 8x8 patches and then reshaped into a sequence of flattened patches. A learnable positional embedding is added to each flattened patch, and a class token (CLS_Token) is added to the sequence. The resulting embedding is then sequentially processed by transformer blocks while also being analyzed in parallel by attention heads, which generate attention filters shown next to each head. The learned representation of the image is adjusted based on a contrastive learning through time loss function.

mounted cameras to capture the child's natural, everyday visual experience [3, 12, 40, 71]. Digital twin studies extend this conceptual approach to newborn animals raised in controlled environments, allowing for tight control over the visual diet available to the animals and machines.

Lee and colleagues [29, 30] used the digital twin approach to compare the learning abilities of newborn chicks and CNNs on a challenging view-invariant object recognition task (Figure 2A). They found that when CNNs were trained "through the eyes" of newborn chicks, the CNNs learned to solve the same view-invariant recognition task as the chicks [57]. Thus, CNNs were not more data hungry than chicks: both could learn robust object features from training data of a single object.

One possibility is that CNNs can learn robust object features from impoverished visual environments because CNNs have a strong architectural inductive bias. CNNs' convolutional operations reflect the spatial structure of natural images, including local connectivity, parameter sharing, and hierarchical structure [28]. This architecture is directly inspired by neurophysiological observations taken from biological visual systems, including a restricted connectivity pattern that resembles the receptive field organization found in the primate visual cortex [16, 24, 69]. CNNs' spatial inductive bias allows them to generalize well from small datasets and learn useful feature hierarchies that capture the structure of visual images. This spatial bias might also explain why CNNs are not more data hungry than newborn chicks (i.e., there is a strong inductive bias supporting visual learning).

ViTs do not have this spatial bias. Rather, ViTs learn through flexible (learned) allocation of attention that does not assume any spatial structure. Thus, ViTs are less constrained by a spatial inductive bias. This flexibility allows ViTs to learn more abstract and generalizable features than CNNs, but it might also make ViTs more data hungry than newborn brains. Indeed, transformers are often regarded as "data-driven learners" that rely heavily on large-scale datasets for training [21, 32, 35, 54].

However, it is an open question whether ViTs are more data hungry than newborn visual systems. For example, [6] report that is it possible to train ViTs with limited data, indicating that ViTs might not be as data hungry as previously thought. That said, the consensus in the field remains that ViTs are data hungry models, especially relative to CNNs [27].

To directly test whether ViTs are more data hungry than newborn visual systems, we performed digital twin experiments on two ViT algorithms. First, we developed a new self-supervised ViT algorithm (**Vi**sion **T**ransformer with **Co**ntrastive Learning through **T**ime: "ViT-CoT") that learns

by leveraging the temporal structure of natural visual experience, using a time-based contrastive learning objective (Figure 2B). ViT-CoT learns without image labels (like newborn chicks) and without artificial image augmentations. The algorithm (Figure 2C) contrasts temporally adjacent instances (positive examples) against non-adjacent instances (negative examples), thereby learning representations that capture the underlying dynamics, context, and patterns across time. We used this contrastive learning through time approach because it has shown promising results with CNN architectures [1, 44] and because there is extensive evidence that biological vision systems leverage time to build enduring object representations [11, 31, 37, 38, 51–53].

Second, to investigate whether our results would generalize across different ViT algorithms, we also trained and tested Video Masked Autoencoders (VideoMAE) [15, 23, 49]. VideoMAE (Figure 4A) contains an encoder-decoder architecture and learns temporal correlations between consecutive frames. VideoMAE encodes spatial and temporal dependencies by masking a subset of space-time patches in the training data and learning to reconstruct those patches. Following prior work, we used a high (90%) masking ratio.

Both ViT algorithms use time as a teaching signal. This is important because newborn chicks rely heavily on temporal information when parsing objects from backgrounds, binding color and shape features into integrated object representations, building view-invariant object representations, and building view-invariant face representations [36, 60–62, 56, 68]. Newborn chicks, like ViT-CoT and VideoMAE, leverage temporal signals to learn how to see.

## 2 Methods

### 2.1 Animal experiments & stimuli

We used newborn chicks as an animal model because chicks can be raised in strictly controlled environments from the onset of vision. This allowed us to control all of the visual experiences (training data) acquired by the animal: an essential requirement for directly comparing the learning abilities of animals and machines. Moreover, chicks can inform our understanding of human vision because the avian brain has similar cells and circuitry as mammalian brains [20, 25, 26]. Avian brains also have a similar large-scale organization as mammalian brains, including a hierarchy of sensory information processing, hippocampal regions, and prefrontal areas.

We focused on the view-invariant object recognition task and behavioral data reported in Wood [57]. In the study, chicks were hatched in darkness, then raised singly in automated controlled-rearing chambers that measured each chick's behavior continuously (24/7) during the first two weeks of life. The chambers were equipped with two display walls (LCD monitors) for displaying object stimuli. As illustrated in Figure 1 (step 1), the chambers did not contain any objects other than the virtual objects projected on the display walls. Thus, the chambers provided full control over all visual object experiences from the onset of vision.

During the training phase, chicks were reared in an environment containing a single 3D object rotating through a 60° viewpoint range. This virtual object was the only object in the chick's visual environment. We modeled the virtual objects after a previous study that tested for invariant object recognition in adult rats [72]. These objects are well designed for studying view-invariant recognition because changing the viewpoint of the objects can produce greater within-object image differences than changing the identity of the object. As a result, recognizing these objects from novel views requires an animal (or machine) to learn invariant object representations that generalize across large, novel, and complex changes in the object's appearance. The chicks were raised in this environment for 1 week, allowing the critical period on filial imprinting to close.

During the test phase (second week), the chicks were tested on their ability to recognize their imprinted object across novel viewpoint changes. On each test trial, the imprinted object appeared on one display wall (from a novel view) and an unfamiliar object appeared on the opposite display wall. Test trials were scored as "correct" when the chicks spent a greater proportion of time with their imprinted object and "incorrect" when the chicks spent a greater proportion of time with the unfamiliar object. Wood [57] collected hundreds of test trials from each chick by leveraging automated stimuli presentation and behavioral coding. As a result, the study produced data with a high signal-to-noise ratio [67].

The chicks performed well on the task when the object was shown from the familiar view and from all 11 novel views [57]. Despite being raised in impoverished visual environments, the chicks successfully learned view-invariant representations that generalized across substantial variation in the object's appearance. Thus, newborn visual systems can build robust object representations from training data of a single object seen from a limited range of views.

## 2.2   Simulating the training data available to chicks

To mimic the visual experiences of the chicks in the controlled-rearing experiments, we constructed an image dataset (Figure 1, step 2) by sampling visual observations from an agent moving through a virtual controlled-rearing chamber. The virtual chamber and agent were created with the Unity game engine. The agent received visual input (64×64 pixel resolution images) through a forward-facing camera attached to its head. The agent could move forward or backward and rotate left or right. The agent could also move its head along the three axes of rotation (tilt, yaw, and roll) to self-augment the data akin to newborn chicks. We collected 80,000 images from each of the four rearing conditions presented to the chicks. We sampled the images at a rate of 10 frames/s.

## 2.3   Contrastive learning through time in ViTs

Studies of newborn perception indicate that newborn animals learn how to see through self-supervised learning, using time as a teaching signal [36, 60–62, 56, 64, 68]. Thus, to compare the learning abilities of newborn chicks and ViTs, we developed a new self-supervised ViT algorithm (ViT-CoT, Figure 2C) that learns through time-based data augmentations (contrastive learning through time). Specifically, for each condition, we initialized the ViT-CoT architecture with three different seeds. Each seed was trained for 100 epochs using a batch size of 128. Throughout the training phase, each image (64×64 pixels) was transformed into 8×8 patches, which were then flattened into a single vector. A randomly initialized positional embedding and a cls_token were then added to this flattened vector before being passed to the ViT encoder. The randomly initialized positional embedding is a trainable parameter that allows the model to learn and capture the spatial information from the input images upon training. The ViT-CoT loss function was:

$$loss_{z_t} = -\log \frac{\exp(\mathrm{sim}(z_t, z_{t+1})/\tau) + \exp(\mathrm{sim}(z_t, z_{t+2})/\tau)}{\sum\limits_{k=1,k\neq t}^{2N} \exp(\mathrm{sim}(z_t, z_k)/\tau)} \tag{1}$$

where N is the number of samples, $z_t$, $z_{t+1}$, and $z_{t+2}$ are the embeddings of the consecutive frames, and $\tau$ is the temperature parameter. We used the value of 0.5 for the temperature parameter.

To approximate the temporal learning window of biological visual systems, the ViT-CoT models had a learning window of three frames (i.e., 300 ms). This 300-ms learning window is based on the observation that neurons fire for 100-400 ms after the presentation of an image, providing a time window for associating subsequent images [34, 43]. The chicks in Wood [57] were reared in one of four possible environments, so we trained each ViT-CoT in a digital twin of one of these four environments (Figure 1, step 3). Thus, the models and chicks had access to the same training data for learning visual representations.

## 2.4   ViT-CoT Architecture Variation

To explore whether ViT-CoTs of different architecture sizes are more data hungry than newborn chicks, we systematically varied the number of attention heads and layers in the models. We built ViT-CoTs with 1, 3, 6, or 9 attention heads and layers. See SI Table 1 for architecture details.

## 2.5   Temporal learning in VideoMAEs

VideoMAEs consider an extra dimension of time (t) to leverage the temporal relationships between consecutive frames. We built a small VideoMAE encoder-decoder model with 3 attention heads and layers in the encoder while having 1 attention head and a single layer in the decoder. See SI Section 6.4 for training details.

## 2.6 Temporal learning in CNNs (SimCLR-CLTT)

Finally, to directly compare ViTs with CNNs, we also trained and tested CNNs. We used the same contrastive learning through time objective function as the ViT-CoT model. Specifically, we used the "SimCLR-CLTT" CNN algorithm [44] to train 10-layer CNNs under each of the four rearing conditions presented to the chicks and ViTs. See SI Section 6.5 for architecture and training details.

## 2.7 Training Data Variation

A leading theory of biological learning is that newborn animals learn how to see by densely sampling the visual environment and encoding statistical regularities in proximal retinal images [13, 22, 47]. By visually exploring the environment and iteratively adjusting connection weights using a temporal learning window, animals might gradually build accurate representations of distal scene variables (e.g., objects), without needing hardcoded knowledge of how proximal image features change as a function of distal scene variables. To explore whether dense sampling allows ViT-CoTs to learn view-invariant object representations in impoverished visual environments, we systematically varied the number of images used to train the ViT-CoTs. We trained the different-sized architectures (1, 3, 6, or 9 attention heads/layers) on 0 images (untrained), 1.25k images, 10k images, and 80k images. By training different sized ViT-CoTs on varying numbers of images, we could explore whether larger ViT-CoTs are more data hungry than smaller ViT-CoTs in the embodied visual learning contexts faced by newborn chicks. We used 80k images to train the VideoMAEs and CNNs.

## 2.8 Supervised linear evaluation

We evaluated the classification performance of the models using the same recognition task that was presented to the chicks. Task performance was assessed by adding a single fully connected linear readout layer on top of the last layer of each trained backbone and then performing supervised training only on the parameters of that readout layer on the binary object classification task. The linear readout layers were optimized for binary cross-entropy loss.

For the linear evaluation, we collected 11,000 simulated images from an agent moving through the virtual chamber for each of the 24 object/viewpoint combinations (2 objects × 12 viewpoints). The object identities were used as the ground-truth labels.

To evaluate whether the learned representations could generalize across novel viewpoints, we systematically varied the number of viewpoint ranges used to train ($N_{train}$) and test ($N_{test}$) the linear classifiers, while holding the number of training images (11,000) constant. We used two different training and test splits, as described below:

- **$N_{train}$ = 11; $N_{test}$ = 1:** We first divided the dataset into 12 folds such that each fold contained images of each object rotating through 1 viewpoint range. The linear classifiers were cross-validated by training on 11 folds (11 viewpoint ranges, 1,000 images each) and testing on the held-out fold (1 viewpoint range: 1,000 images).
- **$N_{train}$ = 1; $N_{test}$ = 11:** In this condition, we inverted the ratio of the training and validation viewpoints. The linear classifiers were trained using only 1 viewpoint range (with 11,000 images) and tested on the remaining 11 viewpoint ranges (1,000 images per viewpoint).

For each linear classifier condition, transfer performance was evaluated by first fitting the parameters of the linear readout layer on the training set and then measuring classification accuracy on the held-out test set. We report average cross-validated performance on the held-out images not used to train the linear readout layer.

## 3   Results

Figure 3A shows the view-invariant object recognition performance of the ViT-CoTs. We also report the performance of the newborn chicks from [57]. All of the ViT-CoTs performed on par or better than chicks when the linear classifiers were trained on 11 viewpoint ranges ($N_{train}$ = 11). We observed similar performance for the small, medium, and large ViT-CoTs (3, 6, and 9 attention heads/layers, respectively). However, we observed a drop in performance for the smallest ViT-CoT (with a single attention head/layer).

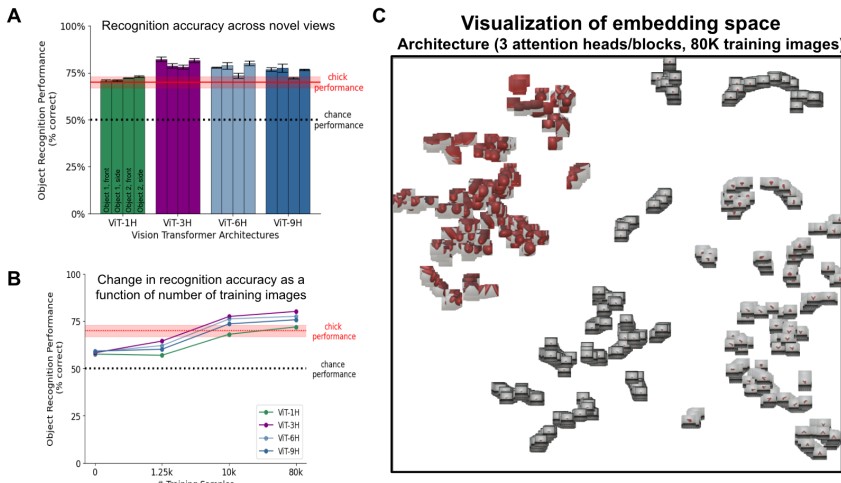

Figure 3: (A) Comparing the view-invariant recognition performance of newborn chicks and ViTs. The red horizontal line shows the chicks' performance, and the ribbon shows standard error. The bars show the view-invariant recognition performance for the four ViT architecture sizes, across the four rearing conditions presented to the chicks. Error bars denote standard error. We used $N_{train}$ = 11 for the linear probe. (B) Impact of the number of training images on view-invariant recognition performance. We trained the ViTs on datasets containing different numbers of images sampled from the virtual chamber. For all architecture sizes, recognition performance improved systematically after training on larger numbers of images, indicating that ViTs can learn view-invariant object representations by densely sampling the visual environment. (C) Two-dimensional t-SNE embedding of the representations (for selected images) in the last layer of a ViT. The features are systematically organized, capturing information about object identity and other important latent variables (e.g., viewing position).

We also found that the number of training images significantly impacted performance (Figure 3B). The untrained ViTs performed the worst, and performance gradually improved when the ViT-CoTs were trained on larger numbers of first-person images from the environment. We observed nearly identical patterns of improvement across the small, medium, and large architecture sizes, indicating that larger ViT-CoTs were not more data hungry than smaller ViT-CoTs.

In the more sparse linear classifier condition ($N_{train}$ = 1), the ViT-CoTs performed on par with the chicks (Figure 1A). This indicates that ViT-CoTs can learn efficient and abstract embedding spaces, which can be leveraged by downstream linear classifiers to support generalization across novel views. Remarkably, the linear classifiers still performed well above chance level even when trained on *a single viewpoint range* (akin to chicks). These results indicate that when ViTs are trained using self-supervised learning, they can form linearly separable and easily accessible view-invariant features. Like chicks, ViTs can learn accessible view-invariant object features in impoverished environments.

The VideoMAE algorithm (Figure 4A) also performed well on the task (Figure 4B), matching the performance of newborn chicks. Thus, our conclusions are not limited to ViT-CoT. Rather, different ViTs show the same learning outcomes, spontaneously developing view-invariant object features when trained on the first-person views available to newborn chicks.

Next, we examined how the CNNs performed on the task (Figure 5). When equipped with a biologically plausible time-based learning objective (CLTT, Figure 5A), the CNNs succeeded on the view-invariant object recognition task (Figure 5B). This finding extends prior work showing that CNNs can solve this task whent they learn from static spatial features [29, 30]. The CNNs did outperform the ViTs (ViT-CoT by 7.3%; VideoMAE by 19.6%), which might be due to the strong architectural inductive bias present in CNNs, but not ViTs. Additional experiments are necessary to determine the specific source of this performance gap. We emphasize that, while ViTs performed lower than CNNs, the ViTs still succeeded on the task, learning view-invariant object features in the same impoverished visual environments as newborn chicks.

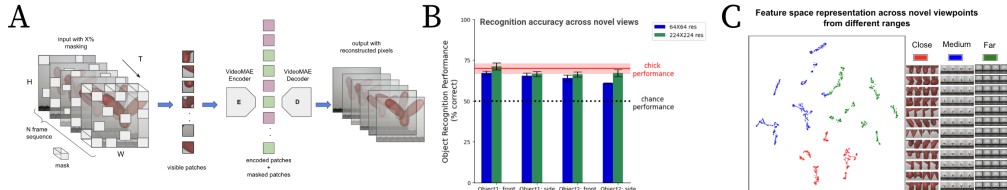

Figure 4: VideoMAEs can learn view-invariant object representations in impoverished visual worlds. (A) VideoMAE architecture with the same transformer encoder as the ViT-CoT model. (B) View-invariant recognition performance of VideoMAE across the four rearing conditions and two image resolution conditions (64x64 or 224x224). We used $N_{\text{train}} = 11$ for the linear probe. The red horizontal line shows the chicks' performance, and the ribbon shows standard error. Error bars denote standard error. ViT-CoT outperformed VideoMAE by 15.8% when trained/tested on 64x64 resolution images and by 6.5% when trained/tested on 224x224 resolution images. (C) t-SNE embedding of the representations in the last layer of a VideoMAE. The features are systematically organized, capturing information about object identity and other important latent variables (e.g., viewing distance shown here).

## 3.1 Controls for image resolution and temporal learning window

To check whether our conclusions generalize beyond the 3-frame learning window, we repeated the ViT-CoT experiments with a 2-frame versus 3-frame learning window. Performance was largely the same across the different learning windows (Figure 3B). To check whether our conclusions generalize beyond the 64x64 image resolution, we repeated the experiments with a higher (224x224) image resolution. Performance was largely the same across the ViT-CoT and Video MAE algorithms (Figure 3C). Finally, we tested whether the models would still succeed on the view-invariant recognition task when both the encoder and the linear classifier were unsupervised. We replaced the supervised linear classifier with an unsupervised decoder. The model scored significantly higher than chance level, which shows that a purely unsupervised learning model, in which both the encoder (ViT) and decoder are trained without any supervised signals, can learn to solve the same view-invariant recognition task as newborn chicks when trained 'through the eyes' of chicks (see SI Section 6.6 for details).

## 3.2 Visualizing ViTs representations

We visualized the features using t-distributed stochastic neighbor embedding (t-SNE), which does not involve using any labeled images. We used t-SNE to visualize the representations in the last layer of the ViT-CoTs (Figure 3C), VideoMAEs (Figure 4C), and CNNs (Figure 5C). The features were systematically organized, capturing information about object identity and other important latent variables (e.g., viewing position). These results are consistent with findings that newborn chicks spontaneously encode information about both the identity and viewpoint of objects [59, 63].

We also created saliency heatmaps from the last layer of the ViT-CoTs (Figure 6). To generate each heatmap, we provided a ViT-CoT with a sample image as input. Then, we determined how much each pixel in the input image produced activation in each attention filter. Finally, for each attention filter, we color-coded each pixel in the input image according to how much it activated the attention filter. Visual inspection indicated that the attention heads became specialized for different features (e.g., whole objects, object parts, floor, depth). The attention heads' specialization varied based on the number of attention heads in the architecture. For models with more attention heads, a single attention head responds to fewer features, compared to a model with fewer attention heads (where each attention head responds to more features). This explains the drop in performance for the single attention head network, which used a single attention head to attend to all the features of the environment. These analyses indicate that ViTs can spontaneously learn enduring visual representations when trained on the embodied data streams available to newborn animals.

# 4 Discussion

We performed parallel controlled-rearing experiments on newborn chicks and self-supervised vision transformers. Using a digital twin approach, we trained chicks and ViTs in the same visual environ-

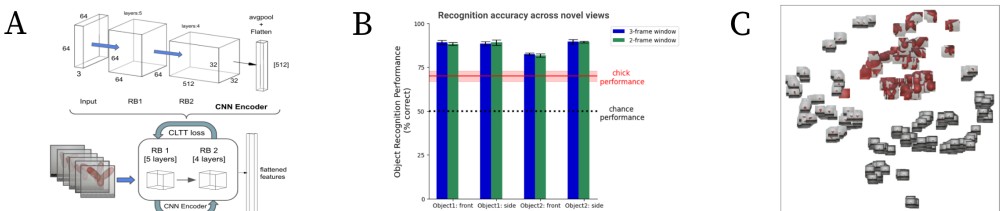

Figure 5: (A) Diagram of SimCLR-CLTT (CNN) model. (B) View-invariant recognition performance of CNNs across the four rearing conditions and two temporal learning windows (2 frames or 3 frames). We used $N_{\text{train}} = 11$ for the linear probe. Error bars denote standard error. The CNNs outperformed ViT-CoT by 7.3%. (C) t-SNE embedding of the representations in the last layer of a CNN trained via contrastive learning through time. The features are systematically organized, capturing information about object identity and other latent variables.

ments and tested their object recognition performance with the same stimuli and task. Our results show that ViTs can spontaneously learn view-invariant object features when provided with the same visual training data as chicks. For both chicks and ViTs, impoverished environments (with a single object) contain sufficient training data for learning view-invariant object features. These results have important implications for understanding minds and machines alike.

## 4.1 Are transformers more data hungry than brains?

A widely accepted critique of transformers is that they are more data hungry than brains. This critique stems from the observation that transformers are typically trained on massive volumes of data, which seems excessive compared to the more sparse experiences available to newborn animals. We suggest that the embodied visual data streams acquired by newborn animals are rich in their own right. During everyday experience, animals spontaneously engage in self-generated data augmentation, acquiring large numbers of unique object images from diverse body positions and orientations. Our results show that self-supervised ViTs can leverage these embodied data streams to learn high-level object features, even in impoverished visual environments containing a single object. Thus, ViTs do not appear to be more data hungry than newborn visual systems, indicating that the gulf between visual learning in brains and transformers may not be as great as previously thought. We support our claim that ViTs are data-efficient learners by showing that two variants of ViTs can both leverage temporal learning to learn like newborn animals.

The field does not have well established procedures for comparing the number of training images across animals and machines, which is why we focused on controlling the visual environment available to chicks and ViTs, rather than controlling the number of training images per se. However, while we cannot ensure that the number of samples is matched between the animals and ViTs, we can make a rough comparison based on the rate of learning in biological systems. Researchers estimate that biological visual systems perform iterative, predictive error-driven learning every 100 ms (corresponding to the 10 Hz alpha frequency originating from deep cortical layers; [39]. If we assume that newborns spend about half their time sleeping, this would correspond to  430,000 images in their first day. Thus, biological visual systems have ample opportunity to learn from "big data."

## 4.2 Origins of object recognition

Our results provide computationally explicit evidence that a generic learning mechanism (ViT), paired with a biologically inspired learning objective (contrastive learning through time), is sufficient to reproduce animal-like object recognition when the system is trained on the embodied data streams available to newborn animals. As such, our results inform the classic nature/nurture debate in the mind sciences regarding the sufficiency of different learning mechanisms for driving biological intelligence. Some researchers have proposed that intelligence emerges from a single generic learning mechanism [33, 41, 48], whereas others have proposed that intelligence emerges from a collection of innate, domain-specific learning systems [7, 46, 10, 17]. Our results show that—for the case of object recognition—a generic learning system (with no hardcoded knowledge of objects or space) is sufficient to learn view-invariant object representations. In fact, ViTs can learn accurate object representations

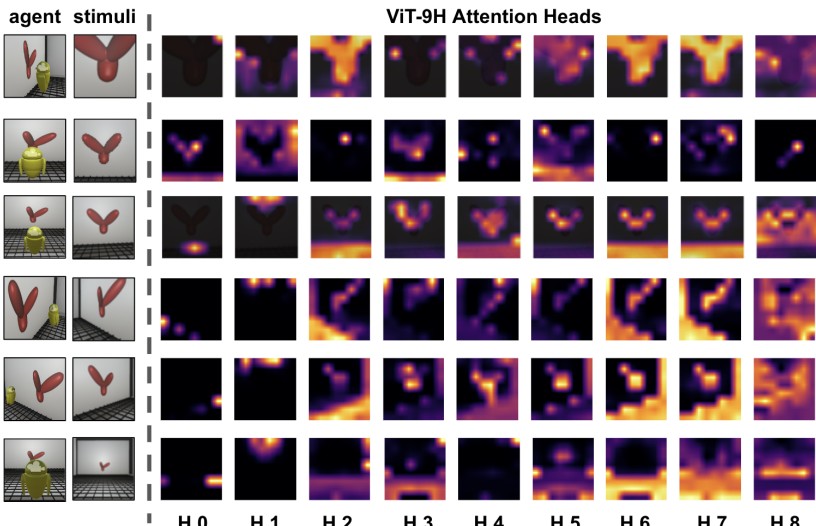

Figure 6: Heatmaps generated from the ViT-9H model. We created saliency heatmaps using the ViT-9H model's last transformer block for all nine attention heads (H0 - H8). The first column shows the agent's position when capturing the first-person image shown in the second column. The subsequent nine columns show the heatmap for each attention head of the ViT-9H model.

even in the impoverished environments faced by chicks in controlled-rearing studies, highlighting both the learning power of ViTs and the rich informational content in embodied data streams.

These results raise an intriguing hypothesis for the origins of visual intelligence: core visual abilities (such as object recognition) might naturally emerge from (1) an attention-based sequence learning system that (2) adapts to the embodied data streams produced by newborn animals. Under this hypothesis, visual intelligence emerges from a single innate sequence learning system that rapidly develops core domain-specific capacities when animals interact with the world during early development. Future research could test this 'single system' hypothesis by exploring whether self-supervised ViTs learn other core visual abilities when they are trained "through the eyes" of newborn animals. Ultimately, parallel controlled-rearing studies of newborn animals and machines can determine which learning mechanisms and experiences are sufficient to produce the rapid and flexible learning of biological systems.

### 4.3 Limitations

One limitation of the current study is that we only tested ViTs across four rearing conditions. Future experiments could test ViTs across a wider range of experimental results, adopting an integrative benchmarking approach used in computational neuroscience [45]. A second limitation of our study is that the models were trained passively, learning from batches of images in a pre-specified order. This contrasts with the active learning of newborn animals, who interact with their environment to produce their own training data. By choosing where to go and what to look at, biological systems generate their own curriculum to suit their current pedagogical needs. Future research could close this gap between animals and machines by embodying ViTs in artificial agents that collect their own training data from the environment.

### 4.4 Broader Impact

This project could lead to more powerful AI systems. Today's AI technologies are impressive, but their domain specificity and reliance on vast numbers of labeled examples are obvious limitations. Conversely, biological brains are capable of flexible and rapid learning from birth. By reverse engineering the core learning mechanisms in newborn brains, we can build "naturally intelligent" learning systems. Naturally intelligent learning algorithms are an untapped goldmine for inspiring the next generation of artificial intelligence and machine learning systems.

# 5 Acknowledgements

Funded by NSF CAREER grant (BCS-1351892, JNW), James McDonnell Foundation Understanding Human Cognition Scholar Award (JNW), and Facebook Artificial Intelligence Research award (JNW).

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

# 6 Supplementary Material

## 6.1 Data generation

To simulate the training and testing data available to chicks, we created a virtual animal chamber in the Unity Game Engine. The virtual chamber was equipped with two 19" LCD monitors situated on opposite sides and accompanied by two alternating white walls. The LCD monitors were used to display the virtual objects of size 8 cm (length) x 7 cm (height), positioned at the center on a white display background. The floor of the virtual chamber was constructed with wire mesh and had transparent holes to hold food and water on one side of the chamber. The virtual chamber measured 66 cm (length), 42 cm (width), and 69 cm (height).

We developed a virtual chick agent to simulate the visual experiences of newborn chicks. The virtual agent had a height of 3.5 units and a length of 1.2 units. To monitor the agent's behavior, the virtual chamber was equipped with two invisible cameras. The first camera, positioned in the chamber's ceiling, captured the top-view of the agent, while the second camera, placed in the agent's head, recorded first-person RGB images (64 x 64 resolution) as the agent moved throughout the chamber.

The agent picked a random location inside the chamber and moved to that position at the rate of 1.5 units/s. During this movement, the agent directed its visual attention towards the object projected on the LCD monitor, ensuring a centered focus. Once the agent reached the destination point, it randomly moved its head along the three axes, rotating 60° on each axis, to naturally augment the data (Figure 1B). This complete random sequence of head rotations lasted for 9.5 seconds. The agent repeated this cycle until they had captured 80,000 first-person images in the rearing condition. We repeated this data simulation cycle for each of the 4 rearing conditions.

The same procedure was repeated to gather the test samples; however, only 11,000 images were captured for each of the 12 viewpoint ranges. These test images were used to train and evaluate the linear classifiers.

## 6.2 Heatmap generation

To visualize the internal working of the attention heads, we generated saliency heatmaps and heatmap videos from the last block of the transformer. Section 3.2 describes the steps required to generate a heatmap for a sample image. To create heatmap videos, we followed these steps. First, the agent moved inside the chamber and captured images at 10 frames/s. Second, we used those images and created saliency heatmaps in a sequential order. Third, we concatenated these heatmaps in a series

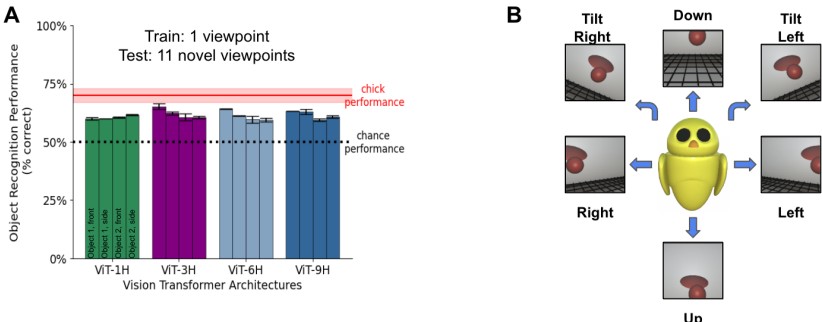

Supplementary Figure 1: (A) Comparing the view-invariant recognition performance of newborn chicks and ViTs in the sparse linear classifier condition (1 training viewpoint). The red horizontal line shows the chicks' performance, with the ribbon representing standard error. The bars show the view-invariant recognition performance for the four ViT architecture sizes, across the four rearing conditions presented to the chicks. Error bars denote standard error. In this sparse linear classifier condition, the linear classifiers are trained on the same viewpoint as the ViT encoder and tested on the remaining 11 novel viewpoints. (B) The virtual agent's head rotations, which provided natural data augmentations. Upon reaching the destination point inside the chamber, the agent randomly rotated its head across the three axes (tilt, left/right, and up/down).

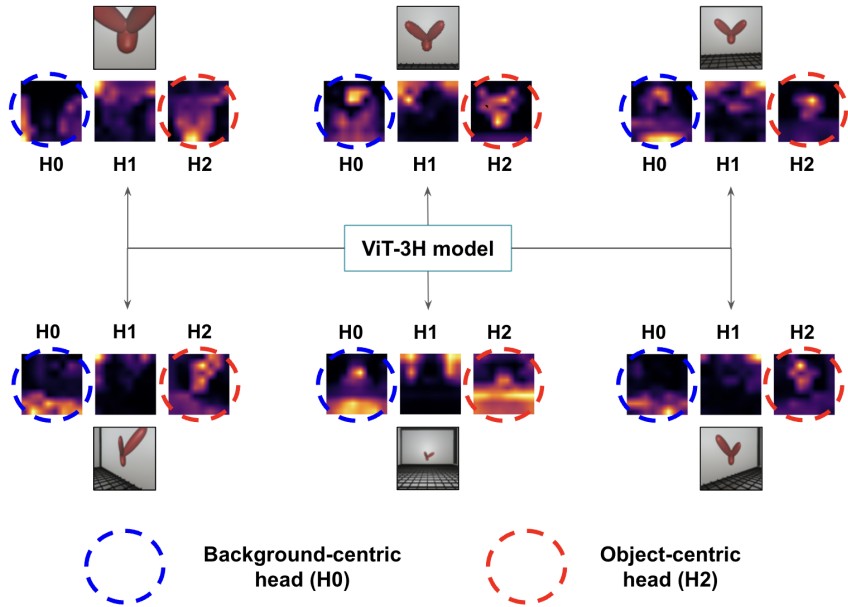

Supplementary Figure 2: Visualizing the attention patterns of individual attention heads using saliency heatmaps derived from the ViT-3H model. The visualizations were generated by using the simulated first-person images captured by the virtual agent. In the ViT-3H model, (H0-H2) correspond to the attention filters of each of the three heads. The blue dotted line highlights that head (H0) largely focuses on background features, while the red dotted line demonstrates that head (H2) largely attends to the object features. This indicates that some attention heads become specialized to different parts of the environment, such as objects versus backgrounds.

Table 1: ViT-CoT with different architecture sizes.

| Model | Parameters | Attention Heads | Transformer Blocks | Batch Size | Epochs |
|-------|-----------|-----------------|--------------------|-----------|--------|
| ViT-1H | 5.8M | 1 | 1 | 128 | 100 |
| ViT-3H | 16.9M | 3 | 3 | 128 | 100 |
| ViT-6H | 36.4M | 6 | 6 | 128 | 100 |
| ViT-9H | 59.4M | 9 | 9 | 128 | 100 |

using the OpenCV library to produce a heatmap video. We repeated these steps for all the attention heads of all the different architectures. Figure 2 displays the heatmaps generated by the ViT-3H model, showcasing the visual representations of the stimulus captured from various angles. Some heads become specialized to the object while other become specialized to the background features.

### 6.3 ViT-CoT Architecture

We systematically increased the number of attention heads and layers to create different architecture sizes ranging from smallest (ViT-1H) to largest (ViT-9H) model (Table 1). The ViT-CoT model contains two primary components: the transformer layers and a SimCLR projection head. The projection head consists of two linear layers and a ReLU activation function. Both of these components are sequentially attached to each other. The projection head receives the output from the final transformer layer as its input and generates an embedding $z$ with a shape of 128. The embedding $z$ is used to adjust the loss function when training the model. During testing, the output is obtained directly from the final transformer layer, without passing through the projection head.

### 6.4 VideoMAE Training

To train the VideoMAEs, we first sampled 16 frames from a batch with a temporal stride of 1. Each frame had a dimension of 64x64 and 3 channels (16x64x64). Next, we randomly masked 90% of the

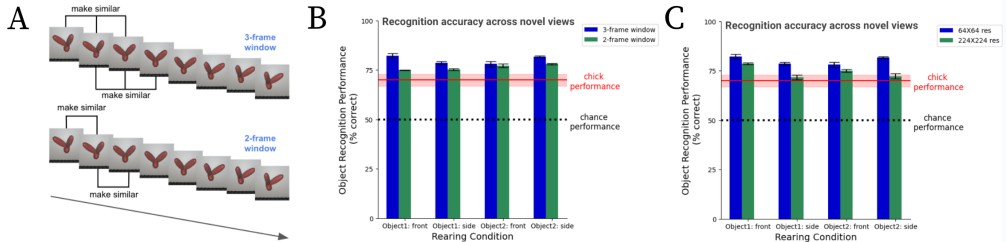

Supplementary Figure 3: (A) Diagram showing the 2-frame versus 3-frame learning windows. (B) View-invariant recognition performance of ViT-CoT across the four rearing conditions and two temporal learning windows (2 frames or 3 frames). Error bars denote standard error. We used $N_{\text{train}} = 11$ for the linear probe. The models with a 3-frame learning window performed marginally better than those with a 2-frame learning window. (C) View-invariant recognition performance of ViT-CoT (3-frame learning window) across the four rearing conditions and two image resolutions (64x64 or 224x224). We used $N_{\text{train}} = 11$ for the linear probe. ViT-CoT performs well with both low and high-resolution images.

frames using cube masks. Each mask had a spatial dimension of 8x8 and a temporal dimension of 2 (2x8x8). The VideoMAE encoder then encoded the visible (non-masked patches) into a latent space with a dimension of 512. Finally, the VideoMAE decoder combined the encoded features with the masked patches to reconstruct the entire sequence of frames. Each VideoMAE model was initialized with three different seeds and trained for 100 epochs using a batch size of 64. We used multi-GPU training across 8 NVIDIA A10 GPUs.

To ensure that our findings with Video MAE would generalize across different image sizes and masking sizes, we repeated this process with a higher image (16x224x224) and larger masks (2x16x16).

### 6.5 CNN Training

We utilized the default ResNet-18 architecture but made it more compact by removing the last two residual blocks, resulting in a 10-layer CNN architecture. We employed 'Contrastive Learning through Time' as the teaching signal, similar to ViT-CoT, for training these CNN models. Each model was initialized with three different seeds and trained for 100 epochs with a batch size of 512.

### 6.6 Two-alternative forced-choice task

In our main experiments, the ViT (encoder) was trained in an unsupervised manner, but we used a supervised linear classifier (decoder) to evaluate the features learned by the ViT. We also tested whether we would obtain similar results when we used an unsupervised, rather than supervised, decoder to evaluate the ViTs. Specifically, we used a modification of the unsupervised technique described by [2], initially developed to test human babies.

The chicks' preferences in [57] can be conceived as a measure of alignment between the test stimuli and the chick's internal representation of their imprinted object. Chicks will approach a stimulus they perceive to be the most similar to their internal representation of their imprinted object (i.e., the stimulus that produces less mismatch, or 'error,' between the stimulus and their representation). To approximate this in silica, we converted each trained ViT model into an autoencoder that was "imprinted" to the same stimulus as the chicks and tested on the same stimuli as the chicks. An autoencoder is trained to reconstruct the original stimulus from a lower dimensional representation, and the reconstruction loss is higher when there is a larger mismatch between a given stimulus and the internal representation.

We converted the models into autoencoders by attaching a simple fully connected downstream decoder to the (trained and frozen) ViT; then we performed unsupervised training on the decoder, using the same images that were used to train the ViT encoder. Consequently, both the encoder and decoder were only trained on images of a single object shown from a single viewpoint range, akin to the chicks. Once the decoder was trained, we used the output from the decoder to quantify how similar each test stimulus was to the ViT's internal representation.

The chicks were tested using a two-alternative forced-choice test (2AFC). To approximate the 2AFC task, we fed two object images into the autoencoder (i.e., ViT + decoder), then measured the error signal for each image. If the error signal was smaller for the imprinted object than the novel object, the model was scored as 'correct.' If the error signal was larger for the imprinted object than the novel object, the model was scored as 'incorrect.' We evaluated the models across all 12 of the viewpoint ranges.

The model scored 62.1%, which is significantly higher than chance level (50%): $X^2(1, N = 576) = 34.03$, $p = .000000005$. This result shows that a purely unsupervised learning model, in which both the encoder (ViT) and decoder are trained without any supervised signals, can learn to solve the same view-invariant recognition task as newborn chicks when trained 'through the eyes' of chicks.

## 6.7 Data Availability

The code and data needed to reproduce these findings can be found at the following link:https://github.com/buildingamind/ViT-CoT.

