# OpenReview forum: "Are Vision Transformers More Data Hungry Than Newborn Visual Systems?"
_NeurIPS.cc/2023/Conference — NeurIPS 2023 poster_

### Official Review · Reviewer_w2HU · 2023-07-05

**Soundness:** 3 good
**Presentation:** 3 good
**Contribution:** 2 fair
**Rating:** 4
**Confidence:** 4

**Summary:**

This work investigates the learning efficiency of vision transformers by comparing their invariant object recognition performance to that of newborn chicks, when being exposed to similar number of images. The authors find that ViTs learn view-invariant representations like chicks and therefore claim that they are not more data hungry than some real animals.

**Strengths:**

The idea behind this paper, which is to train visual models using the same input real organisms can receive and to model the animal development using such models, is a good idea. The paper is also in general well written with clear logic. To confirm that the results are general, the authors tested multiple architectures. The embedding space visualization result shows the potential explanation for why multiple attention-heads outperform single attention-head models.

**Weaknesses:**

The biggest weakness is that the starting point, that ViTs are thought to be more “data-hungry” than brains, is already proven false in earlier works. A paper published last year (2022) on ECCV, which is on arxiv even earlier, shows that ViTs achieve reasonable performance even when trained on 2048 images [1]. Although the authors repeatedly mention that ViTs are more thought to be more “data-hungry”, they did not provide any citation supporting this point. They also did not discuss why this point is still worth being investigated given this earlier work.

In addition to this, the linear classification training used during the test phase is also unjustified in the paper. This training requires supervision signal to be given to the model, but this signal does not seem to be available to the real animal. This makes the interpretation of the test phase results potentially wrong. A much simpler classifier that requires very little or no training should be used in this phase, such as a correlation classifier to earlier stored “imprinted” object hidden representations. The current results may very well be an overestimation of the network’s performance under a simpler classifier.

It is also unclear to me whether the simulated input during the training phase closely reflect the input visual statistics real chicks can receive. The authors designed certain agent movement patterns, is this pattern similar to what real newborn chicks would do? How about the frequency of this “movement cycle”? Is this leading to more or less augmentations compared to the real animal?

[1] Cao, Yun-Hao, Hao Yu, and Jianxin Wu. "Training vision transformers with only 2040 images." European Conference on Computer Vision. Cham: Springer Nature Switzerland, 2022.


**Questions:**

-	Address the earlier works better about whether ViTs are more data hungry than real animals.
-	Replace the supervised linear classification in the test phase with a much simpler classifier that requires no or little training.
-	Does the simulated input during the training phase closely reflect the input visual statistics real chicks can receive?

**Limitations:**

yes

---

> ### Author Rebuttal · Authors · 2023-08-08
>
> Thank you for your feedback and time. We address your 3 critiques below.
>
> >#1: The biggest weakness is that the starting point, that ViTs are thought to be more “data-hungry” than brains, is already proven false in earlier works...
>
> We thank the reviewer for pointing out this citation. This concern highlights an opportunity for us to address this debate in our Related Works section. In the new text, we will discuss that the general consensus in the field is still that ViTs are data hungry models (especially relative to CNNs). For example, a recent (2022) review of vision transformers notes that, “CNNs encode prior knowledge about the images... that reduces the need of data as compared to Transformers that must discover such information from very large-scale data.” (Khan et al., 2022, ACM Computing Surveys). Indeed, even the Cao, Yu, and Wu (2022) paper explains that ViTs, “achieve competitive results with CNNs but the lack of the typical convolutional inductive bias makes them more data-hungry than common CNNs” and that “ViT and variants achieve competitive results with CNNs but require significantly more training data. For instance, ViT performs worse than ResNets with similar capacity when trained on ImageNet (1.28 million images). One possible reason may be that ViT lacks certain desirable properties inherently built into the CNN architecture that make CNNs uniquely suited to solve vision tasks, e.g., locality, translation invariance and hierarchical structure. As a result, ViTs need a lot of data for training, usually more data-hungry than CNNs.”
>
> We will also point readers to work that has tried to reduce ViTs’ dependence on large scale training data. For example, some researchers have proposed adding a CNN architecture to ViTs to take advantage of the spatial inductive bias inherent in CNNs (e.g., Yuan et al., 2021, ICCV). Another approach from Cao et al. (2022) is to introduce pre-training with artificial augmentations to the dataset (e.g., multi-crop and CutMix).
>
> Critically, however, our approach differs in 3 critical ways from Cao et al. (2022). First, we used a single visual object in our training set with no extra data augmentation, whereas Cao et al. used 2040 images of many different objects with extra data augmentation. Thus, our results demonstrate that ViTs are even less data hungry than previously shown. Second, we used biologically plausible data augmentations that were generated by simulating the visual experiences of an agent moving through virtual replicas of the animal chambers. In contrast, Cao et al. (2022) used artificial augmentations like CutMix (an augmentation that would be impossible for an actual animal to perform). Third, our claim is that ViTs are not more data hungry than newborn visual systems. The only way to address this claim directly is to train ViTs and newborn animals with the same tasks and test ViTs and newborn animals with the same tasks. In this sense, our findings are unique from any existing work on ViTs (including Cao et al., 2022). Moreover, our results have important scientific implications. ViTs have the potential to be powerful image-computable models of newborn visual systems, but these models will not be accepted if they are thought to be more data hungry than brains. Our results directly contradict this widely held assumption, and thus, will shape our growing understanding of the relationship between brains and transformers.
>
> >#2: ...A much simpler classifier that requires very little or no training should be used in this phase, such as a correlation classifier to earlier stored “imprinted” object hidden representations. The current results may very well be an overestimation of the network’s performance under a simpler classifier.
>
> We used supervised linear classifiers to evaluate the unsupervised ViTs because this is the standard approach in computational neuroscience for quantifying the performance of self-supervised models. While supervised learning was not present in the chick experiments, linear classifiers are simply a formal way of quantifying the degree and form of learned representations. In neuroscience, information that is available directly via a linear readout is generally considered to be explicitly represented by a model or brain region. The linear classifier does not provide the ViT with new information but merely measures the relative placement of different images within the model’s existing feature space. Linear classifiers are also a reasonable approximation of downstream neural computation, since linear classifiers express a plausible rate-code model for downstream decoder neurons (i.e., linear weightings followed by a single threshold value; Hong et al., 2016, Nature Neuroscience).
>
> Accordingly, the most common way to evaluate unsupervised models that are trained on biologically plausible training data (e.g., simulated first-person images from chick experiments) is to test the unsupervised model’s ability to classify objects using a supervised linear readout (c.f., Zhuang et al., 2022, NeurIPS; Zhuang et al., 2021, PNAS; Orhan, Gupta, & Lake, 2020, NeurIPS).
>
> >#3: It is also unclear to me whether the simulated input during the training phase closely reflect the input visual statistics real chicks can receive. The authors designed certain agent movement patterns, is this pattern similar to what real newborn chicks would do? How about the frequency of this “movement cycle”? Is this leading to more or less augmentations compared to the real animal?
>
> We designed our agents to use the same six degrees of freedom for head movements (roll, pitch, yaw) as newborn chicks. However, it is not possible to affix cameras to a chick’s head, so our simulations may not reflect the movement cycles of the animals. In future research, we plan to characterize the nature of chicks’ actual visual statistics by using tools like DeepLabCut and Unity to yoke the virtual agent to actual chicks during the Input Phase of the experiment.

---

> > ### Comment · Reviewer_w2HU · 2023-08-16
> >
> > Thanks for the rebuttal. The authors addressed my first question well, but the response to my other two questions are not convincing to me.
> >
> > For the second question, what the authors want to achieve in this work is a strict comparison with the performance from the real chicks, which was measured in an earlier work using a specific testing method. I don’t think the linear classifier training is possible under that testing method, as training the classifier requires a lot of supervision, which simply does not exist in that testing phase. Although linear classifier training was used in earlier works, it does not mean that it can be used here as the authors attempt to compare the performance of the networks to the animals. Also, the final difference between the models and the animals is small. So, it’s highly possible that how this readout method is done is critical to this comparison.
> >
> > For the third question, can the authors show how this “cycle frequency” influences the performance? If the performance is not influenced significantly by this cycle frequency parameter across a large range of values (especially for lower values), I am more convinced that the current result is not just due to a specific value for this parameter.
> >
> > I will increase my score to 4 right now and am willing to further increase the score if the authors can address my questions here well.

---

> > > ### Author Response · Authors · 2023-08-19
> > >
> > > >Can the authors show how this “cycle frequency” influences the performance?
> > >
> > > We thank the Reviewer for clarifying their critique regarding “cycle frequency.” In prior experiments with CNNs, we tested whether different parameters of cycle frequency (i.e., different amounts of data augmentation due to movement patterns) significantly impact performance. Specifically, in a “no head rotation” condition, the agent collecting the images stared continuously at the object as they moved around the chamber, so the object was always in the middle of the camera. Conversely, in the “head rotation” condition, the agent rotated their head 30° in each direction along the 3 axes of rotation (yaw, pitch, roll) in a random order while looking at the object. Consequently, the agent in the head rotation condition collected many more unique views of the object than the agent in the no head rotation condition, due to the data augmentation produced by the head rotations. We found that view-invariant object recognition performance was nearly identical across these two conditions, so we focused on the “head rotation” condition for the present ViT experiments.
> > >
> > > We are, however, in the process of adding a “no head rotation” condition to the present paper testing ViTs to address this concern.
> > >
> > > >What the authors want to achieve in this work is a strict comparison with the performance from the real chicks, which was measured in an earlier work using a specific testing method. I don’t think the linear classifier training is possible under that testing method, as training the classifier requires a lot of supervision, which simply does not exist in that testing phase. Although linear classifier training was used in earlier works, it does not mean that it can be used here as the authors attempt to compare the performance of the networks to the animals.
> > >
> > > We agree with the Reviewer: a direct test of the learning abilities of newborn chicks and ViTs will ultimately require an entirely unsupervised training/testing approach, since the chicks were trained/tested without any supervisory signals. In our original submission, the ViT (encoder) was trained in an unsupervised manner, but we used a supervised linear classifier (decoder) to evaluate the features learned by the ViT. Following the Reviewer’s suggestion, we will add new experiments to the paper that use unsupervised decoders to evaluate the ViTs. Specifically, we used a modification of the unsupervised technique described by Ayzenberg & Lourenco (2022_eLife), initially developed to test human babies.
> > >
> > > The chicks’ preferences in Wood (2013) can be conceived as a measure of alignment between the test stimuli and the chick’s internal representation of their imprinted object. Chicks will approach a stimulus they perceive to be the most similar to their internal representation of their imprinted object (i.e., the stimulus that produces less mismatch, or ‘error,’ between the stimulus and their representation). To approximate this in silica, we converted each trained ViT model into an autoencoder that was “imprinted” to the same stimulus as the chicks and tested on the same stimuli as the chicks. An autoencoder is trained to reconstruct the original stimulus from a lower dimensional representation, and the reconstruction loss is higher when there is a larger mismatch between a given stimulus and the internal representation.
> > >
> > > We converted the models into autoencoders by attaching a simple fully connected downstream decoder to the (trained and frozen) ViT; then we performed unsupervised training on the decoder, using the same images that were used to train the ViT encoder. Consequently, both the encoder and decoder were only trained on images of a single object shown from a single viewpoint range, akin to the chicks. Once the decoder was trained, we used the output from the decoder to quantify how similar each test stimulus was to the ViT’s internal representation.
> > >
> > > The chicks were tested using a two-alternative forced-choice test (2AFC). To approximate the 2AFC task, we fed two object images into the autoencoder (i.e., ViT + decoder), then measured the error signal for each image. If the error signal was smaller for the imprinted object than the novel object, the model was scored as ‘correct.’ If the error signal was larger for the imprinted object than the novel object, the model was scored as ‘incorrect.’
> > >
> > > The model scored 62.1%, which is significantly higher than chance level (50%): *X*2(1, N = 576) = 34.03, *p* = .000000005. This new result shows that a purely unsupervised learning model, in which both the encoder (ViT) and decoder are trained without any supervised signals, can learn to solve the same view-invariant recognition task as newborn chicks when trained ‘through the eyes’ of chicks.
> > >
> > > We hope this fully satisfies all of the remaining concerns. We thank the Reviewer for encouraging us to pursue the unsupervised learning experiments: we think they greatly improve the paper!

---

### Official Review · Reviewer_Au5T · 2023-07-05

**Soundness:** 3 good
**Presentation:** 3 good
**Contribution:** 4 excellent
**Rating:** 7
**Confidence:** 5

**Summary:**

This study challenges the notion that Vision Transformers (ViTs), which excel in many computer vision benchmarks, require more training data than biological brains. The study involved controlled experiments on both ViTs and newborn chicks in impoverished visual environments, using a video game engine to simulate the chicks' environments and train self-supervised ViTs that use time as a teaching signal, similar to biological systems. The authors found that when trained in conditions similar to those of newborn chicks, ViTs effectively performed the same object recognition tasks, suggesting that ViTs are not necessarily more "data hungry" than biological systems and can develop animal-like object recognition capabilities through generic attention-based learning mechanisms.

**Strengths:**

The main strength of this paper is the novelty of training chicks with a regimented visual schedule and ensuring that agents are trained the same way for fair comparison. It is a sorely needed paradigm for claims of data efficiency during organism developmental timescales.

**Weaknesses:**

There were a couple clarifications regarding the experimental setup and model comparison that can be clarified in the camera ready. I have listed my specific questions in the next section below.

**Questions:**

1.	In line 233, it is stated that the “embodied visual data streams acquired by newborn animals are rich in their own right.” How do you ensure that the agent visual stream is matched to that of the animal in terms of number of samples?
2.	In lines 111-112, it is said that “the study produced data with a high signal-to-noise ratio”. What does this mean exactly, and how is it quantified? What would noise be in this context?
3.	In line 122, it is stated that “The agent received visual input (64x64 resolution images)”. Is this matched physically to the visual acuity of newborn chicks? If not, what would the appropriate resolution be, and what happens when you train ViTs at that resolution?
4.	In line 125, it is mentioned there are 4 rearing conditions. What are they? I didn’t seem to see that in the main text but could have mistakenly missed it.
5.	Is Figure 3A on heldout test performance, where Ntest = 1? Please clarify in the figure caption.
6.	Can you make any statements or predictions about data efficiency of chicks when there is more than one object to be learned? (cf. lines 234-235).
7.	Minor: How do temporally-contrastive CNNs match ViTs in this context in terms of number of training samples? It would be good to have some architectures other than ViT to show whether adding more inductive biases increases or decreases training efficiency.


**Limitations:**

Yes, the authors have adequately addressed the limitations of their work.

---

> ### Author Rebuttal · Authors · 2023-08-08
>
> Thank you for the positive feedback. We address your questions below:
>
> > In line 233, it is stated that the “embodied visual data streams acquired by newborn animals are rich in their own right.” How do you ensure that the agent visual stream is matched to that of the animal in terms of number of samples?
>
> Great question. Ultimately, the field does not have well established procedures for comparing the number of training images across animals and machines, which is why we focused on controlling the visual environment available to chicks and ViTs, rather than controlling the number of training images per se. However, while we cannot ensure that the number of samples is matched between the animals and ViTs, we can make a rough comparison based on the rate of learning in biological systems. Researchers estimate that biological visual systems perform predictive error-driven learning every 100 ms (corresponding to the 10 Hz alpha frequency originating from deep cortical layers; O’Reilly et al, 2021). If we assume that newborns spend about half their time sleeping, this would correspond to ~430,000 images in their first day.
>
> We also emphasize that a widely accepted critique of ViTs is that they are more data hungry than animals. This critique stems from the observation that ViTs are typically trained on millions of images across thousands of object categories, which seems excessive compared to the visual environments of newborn animals. Our paper shows that the embodied data streams acquired by newborn animals are rich in their own right. During everyday experience, newborn animals spontaneously engage in self-generated data augmentation, acquiring large numbers of unique object images from diverse body positions and orientations. We show that ViTs, like animals, can leverage these embodied data streams to learn high-level object features in impoverished visual environments (which we will clarify in the Camera Ready).
>
> > In lines 111-112, it is said that “the study produced data with a high signal-to-noise ratio”. What does this mean exactly, and how is it quantified? What would noise be in this context?
>
> By “noise,” we mean unexplained inter-subject variation, which we can measure as the standard deviation of performance between chicks. By “signal-to-noise ratio,” we mean the size of the effect (the mean difference between chick performance and chance performance) compared to the variability. We quantify the signal-to-noise ratio as Cohen’s d (the “standardized mean difference” = mean difference / standard deviation). Our revision will point readers to Wood & Wood (2019) for a detailed explanation of how this method produces precise measurements of performance with large effect sizes.
>
> > In line 122, it is stated that “The agent received visual input (64x64 resolution images)”. Is this matched physically to the visual acuity of newborn chicks? If not, what would the appropriate resolution be, and what happens when you train ViTs at that resolution?
>
> Great question! Chick visual acuity is about 25% of human visual acuity, which is why we initially used lower resolution images. However, we also performed new experiments to test whether different image resolutions would impact our results. To do so, we trained both the ViT-CoT and VideoMAE models with 224x224 resolution images. We did not observe a large difference between the small (64x64) and large (224x224) image resolutions, so we do not believe that image resolution strongly impacted our results. We will add these new experiments to our Camera Ready version.
>
> > In line 125, it is mentioned there are 4 rearing conditions. What are they? I didn’t seem to see that in the main text but could have mistakenly missed it.
>
> In the chick experiments, each chick was raised with one of two objects (Object 1 vs. Object 2) presented from one of two viewpoints (front vs. side), making 4 rearing conditions in total. We will add this information to the Supplemental Materials.
>
> > Is Figure 3A on heldout test performance, where Ntest = 1? Please clarify in the figure caption.
>
> Yes, in Figure 3A, performance is held out, cross-validated test performance where Ntest = 1. (The results for Ntest = 11 are in the Supplemental Materials.) We will make this explicit in the Camera Ready version.
>
> > Can you make any statements or predictions about data efficiency of chicks when there is more than one object to be learned?
>
> This is a great question. At this point, we cannot make any statements about the data efficiency of chicks when there is more than one object to be learned, since all of our chick experiments have focused on how chicks learn their first object representation. In the near future, we will start exploring how chicks learn multiple objects, potentially providing data for distinguishing between candidate image-computable models of newborn visual systems.
>
> > How do temporally-contrastive CNNs match ViTs in this context in terms of number of training samples? It would be good to have some architectures other than ViT to show whether adding more inductive biases increases or decreases training efficiency.
>
> We agree that a comparison between CNNs and ViTs would be valuable for determining how different inductive biases may impact the development of vision. To do so, we added new experiments comparing CNNs with ViTs, in which we evaluated a temporally contrastive CNN architecture (SimCLR-CLTT) using the same training and test conditions we used for the ViTs (ViT-CoT in original submission and new VideoMAE results added during rebuttal). We found two interesting patterns: (1) both CNNs and ViTs could solve the task (i.e., learn view-invariant object representations from impoverished visual environments) and (2) the stronger inductive biases of the CNNs led to a small but significant bump in performance over ViTs. These findings suggest that starting with a CNN architecture is beneficial, but not necessary, for learning view-invariant object representations.

---

> > ### Comment · Reviewer_Au5T · 2023-08-14
> > **Thank you!**
> >
> > Thank you for your thorough & detailed response, including the additional experiments. I find the results with the CNN especially intriguing, and I am glad you will be including it in the revision. I hope to see this paper accepted, and I will be advocating for it!

---

### Official Review · Reviewer_xTn4 · 2023-07-07

**Soundness:** 2 fair
**Presentation:** 3 good
**Contribution:** 2 fair
**Rating:** 4
**Confidence:** 4

**Summary:**

This article examines the issues of data-hungry between chicks and ViT. For the sake of experimental accuracy, this paper makes a dark room and controls some variables in terms of organisms. In terms of ViT, ViT-CoT is proposed, and data enhancement, embedding and simple modification of the model are carried out. Finally, a comparative experiment was carried out, which proved the experimental results well.

**Strengths:**

Clear writing and good figures for easy understanding.
Interesting perspective on comparing newborn and vit.


**Weaknesses:**

There is a big difference between living organisms and computers. So that some settings in the experiment are not enough to rule out the influence of other aspects. Conclusions are somewhat subjective. For example, the newborn chicken is designed by humans and implemented by human, and as far as I learn from this paper, this design can only imitate limited features of living organisms. Also, why chicken can be represented for the “newborn visual system”.

Inadequate verification of the appropriateness of the task for addressing the topic issue. As far as I learn from this paper, the final task for ViT and newborn system is to recognize object across novel views, and the objects are composed of virtual and designed geometry element. The difficulty and appropriateness of this task should be studied thus the conclusion can be convinced.

This paper proposes a new contrastive loss function to train ViT as self-supervised learning. However, the other powerful self-supervised method, i.e., mask some part of the image and train the model to reconstruct the complete image, is not discussed and tested. This leads to insufficient method design and experiment conduction, since the subject of this paper gives no limitation to the training method.


**Questions:**

On the (129 line, 4 page) “Thus, to compare the learning abilities of newborn chicks and ViTs, we developed a new self-supervised ViT algorithm...... Specifically, ViT-CoT architecture was initialized with three different seeds ……”. This paper uses time-based data augmentations and studies spatial information from the input images. This paper adds a lot of advantages to the training of vit, but the chick does not. For example, when a chick learns a three-dimensional model of an object, it may be naturally strongly correlated with its spatial variation. The chicks may be confused when it sees three-dimensional objects move but their own position in space does not change.

We all know that the difference between living organisms and computers is that living organisms may encounter fatigue, illness, or other bad conditions during the learning process, resulting in a decline in learning ability. Even without these interfering factors, it still needs to grow and eat by itself. And it is disturbed by the smell, temperature, feather movement, etc. How to control or reduce the occurrence of such problems in this experiment? If the learning ability of organisms is underestimated because of such problems, how to draw that conclusion of this paper?

In Figure 3, why does the accuracy of the chick not increase with the training time?


**Limitations:**

The paper lacks discussion on the setting of the proposed task and effects of other training methods.

---

> ### Author Rebuttal · Authors · 2023-08-09
>
> Thank you for your feedback and time. We address your critiques below.
>
> *Reviewer raises two concerns: (1) Why use chicks as a model system for studying newborn vision? (2) Our design can only address some features of animals.*
>
> For (1), our revision will clarify why chicks are optimal for studying newborn visual learning. We used chicks as a model system because chicks can be raised in controlled environments from the onset of vision. This allows us to control all of the visual experiences (training data) acquired by the animal: an essential feature for directly comparing the learning abilities of animals and machines. Moreover, chicks can inform our understanding of human vision because the avian brain has similar cells and circuitry as mammalian brains (Güntürkün & Bugnyar, 2016; Jarvis et al., 2005; Karten, 2013). Avian brains also have a common large-scale organization as mammalian brains, including a hierarchy of sensory information processing, hippocampal regions, and prefrontal areas.
>
> For (2), we agree that our study only tests specific learning abilities of animals, but this is an essential first step in this research program. Ultimately, we introduce a powerful new experimental approach (the first of its kind) for directly comparing the learning abilities of animals and transformers. It would be interesting for future research to explore whether ViTs can replicate other features of living organisms (e.g., object parsing, face recognition, motor development, navigation, and audition).
>
> *Reviewer argues that we did not verify the appropriateness of the task for addressing the topic issue.*
>
> Our revision will clarify the appropriateness of the task. We modeled the virtual objects and task after a previous study that tested for invariant object recognition in adult rats (Zoccolan et al., 2009). The objects are well designed for studying view-invariant recognition because changing the view of each object produces a greater within-object image difference than changing the identity of the object. Thus, recognizing these particular objects from novel views requires an animal (or model) to learn abstract object features that can generalize across large, novel, and complex changes in the object’s appearance.
>
> *Reviewer argues that we had an insufficient method design because we did not test other ViTs that use masking approaches.*
>
> We agree that MAEs are an important candidate for unsupervised visual learning. As suggested, we performed new experiments with the VideoMAE model. We trained/tested VideoMAE using the same approach that we used to train/test ViT-CoT. We found that VideoMAE, like ViT-CoT, can perform the task, showing that vision transformers are sufficient to drive the development of animal-like object recognition.
>
> *Reviewer argues that the temporal learning approach is appropriate for ViTs, but not for chicks.*
>
> The reviewer is concerned that ViT-CoT uses time-based data augmentation, but that chicks instead use learning mechanisms that are correlated with their own position in space. We will revise the manuscript to emphasize that chicks are highly sensitive to temporal information, thereby justifying our use of models that leverage temporal information to learn. For example, newborn chicks use temporal information when parsing objects from backgrounds (Wood & Wood, 2021a), binding color and shape features into integrated object representations (Wood, 2016), building view-invariant object representations (Wood & Wood, 2016; 2018), and building view-invariant face representations (Wood & Wood, 2021b).
>
> *Reviewer argues that animals have needs (e.g., hunger, fatigue) that are not present in the ViT experiments, making comparisons across chicks and ViTs difficult.*
>
> We agree; it is inevitable that any comparison between humans/animals and machines will open the possibility for fatigue, illness, hunger, etc. to lead to noisy estimates of biological learning. To minimize the impact of these factors, our design uses automation and long test periods. As reviewed in Wood & Wood (2019), this methodology produces data with a high signal-to-noise ratio and strong test-retest reliability across individual chicks.
>
> We also note that, while many factors may contribute to differences between animals and machines, the critical factor needed to compare learning across animals and machines is that both are provided with the same training data and tested on the same tasks. Currently, no other method comes close to achieving this goal. Accordingly, while other benchmarks can be used to evaluate whether trained models behave like mature animals, these benchmarks cannot be used to determine whether models use similar learning algorithms as animals (because the animals and models learned from different training data). We therefore argue that a major contribution of our paper is to add a new paradigm to the field that can unambiguously reveal whether animals and machines learn in the same way.
>
> *Reviewer wonders why chick performance did not increase across the training period.*
>
> Thank you for pointing out the lack of clarity. Fig. 3 shows the performance of ViT-CoTs (not chicks) trained on various numbers of training samples. The red line showing chick performance is provided so readers can easily see the number of training samples needed to reach chick-level performance. We will clarify this in the revision.
>
> *Reviewer argues that the paper lacks discussion on the setting of the proposed task and effects of other training methods.*
>
> To make the setting of the task clearer, we will direct readers to Wood (2013) for a detailed description of the task. Moreover, to provide a new dataset to the community (and to help clarify the setting of the machine task), we will provide the full set of training and test images that we simulated in the “digital twin” experiments. Finally, to address the effects of other training methods, we also added new experiments using VideoMAEs and CNNs (see rebuttal PDF).

---

> ### Comment · Area_Chair_ZJ86 · 2023-08-18
>
> Dear Reviewer xTn4,
>
> We are nearing the end of the discussion period with authors.
>
> The authors have responded in detail to your review, so pls minimally read and acknowledge their rebuttal, and state which (if any) issues you still do not find to be satisfactorily addressed.
>
> You should do so as soon as possible.
>
> Thanks,
> AC

---

### Official Review · Reviewer_8hgW · 2023-07-07

**Soundness:** 3 good
**Presentation:** 4 excellent
**Contribution:** 3 good
**Rating:** 5
**Confidence:** 4

**Summary:**

The authors present a study showcasing, in one specific scenario, a vision transformer can match the visual learning performance of newborn chicks. The setup is as follows. Newborn chicks are raised in a dark enclosure for a week, and given only one visual stimulus from a variety of angles. Then, the chicks are presented with the visual stimulus from new angles, along with different visual stimuli. The chicks show a preference towards the same stimulus from new angles, which means that they have learned an angle-invariant generalization about the stimulus. Vision transformers are shown the same stimulus in a simulated environment, are trained on the simulated images, and are probed for the ability to learn angle-invariant generalizations too.

**Strengths:**

Although not entirely novel (similar experiments have been done with CNNs and similar results have been shown with them), I do think that this is interesting work, and I did not find any issues with the experimental setup or results. It extends the previous findings from CNNs to transformers to showcase that transformers do not require more data than a newborn animal to learn the same generalization in one specific scenario.

**Weaknesses:**

Not sure I really buy the 3-frame learning window idea. Does the algorithm actually not work if you only associate two frames with each other instead of three?

Do you say how many parameters your vision transformers have anywhere? This would be really useful to know.

Instead of a simulator, why not run the test with an actual robot driving/walking around? It would make the work more novel and interesting in my opinion.

**Questions:**

I tried to put my points in "weaknesses" as questions.

**Limitations:**

Limitations are adequately addressed in my opinion.

---

> ### Author Rebuttal · Authors · 2023-08-08
>
> Thank you for your valuable questions and feedback, which we address below:
>
> > Not sure I really buy the 3-frame learning window idea. Does the algorithm actually not work if you only associate two frames with each other instead of three?
>
> We agree that the learning window duration is an interesting hyperparameter. We chose a learning window of 3 frames to approximate the temporal learning window in biological visual systems (around 300ms, which corresponds to 3 of our simulated frames). As the Reviewer points out, the learning window may influence the algorithm’s ability to learn visual features. To test this, we replicated our experiments with a 2-frame learning window. We find that the 2-frame learning window also performs well on the task, demonstrating that the algorithm’s success is not dependent on a learning window of 3 frames.
>
>
> > Do you say how many parameters your vision transformers have anywhere? This would be really useful to know.
>
> Our vision transformers have 5.8M (1-head architecture), 16.9M (3-head architecture), 36.4M (6-head architecture), and 59.4M (9-head architecture) trainable parameters, respectively. We provide details on the number of trainable parameters in Table 1 of the Supplementary Materials.
>
>
> > Instead of a simulator, why not run the test with an actual robot driving/walking around? It would make the work more novel and interesting in my opinion.
>
> This is an excellent idea for future research! In the present study, we focused on whether ViT models can account for the visual processing abilities of newborn animals (which is why we focus on “disembodied” models of vision that do not produce actions). However, we are currently building a platform for rearing embodied virtual agents in “digital twins” of the controlled rearing chambers, which will allow us to train real and artificial animals with the same data and test them with the same tasks.
>
> We currently use virtual agents (rather than robots) because of their scalability. First, we can run virtual simulations with dozens of agents simultaneously, at faster than real-time speeds. As a result, we can collect more data and test more models. Second, using virtual agents allows our work to scale beyond our lab. We can make “digital twins” of our chambers available to researchers outside of our lab, so that they can replicate our findings and/or test new models. Finally, once we discover core learning algorithms that develop like newborn animals, we will plug our virtual brains into physical robots and test whether policies learned virtually generalize to the real world.

---

> > ### Comment · Reviewer_8hgW · 2023-08-16
> >
> > Thanks for the detailed responses to my concerns - I think that your paper would benefit from including the information that you responded with!
> >
> > I still like the paper and my rating still leans slightly in the accept direction. Although, from reading your rebuttal it seems that you are planning to revise your paper to highlight these items:
> >
> > * the visual twin approach is novel.
> >
> > * ViTs are not more data hungry in general.
> >
> > I think that you will face some criticism for making these claims, and I would advise against the wording in your rebuttal. From my understanding, the visual twin approach is not entirely novel because it was done with CNNs in prior work. And, yes, I would say that you've provided some evidence that ViTs are not more data hungry, but only in a very specific case. My rating might lean slightly in the reject direction if you make the claims in the paper too strong.

---

### Author Rebuttal · Authors · 2023-08-08

Our paper tackles a question at the heart of biological and artificial intelligence: Are vision transformers (ViTs) more data hungry than newborn visual systems? The answer to this question will have significant implications on (1) how transformers are viewed by AI researchers (e.g., brain-like or not brain-like?) and (2) whether scientists embrace transformers as viable models of biological visual systems. The dominant assumption is that ViTs are more data hungry than visual systems, but we showed that this assumption is incorrect. To demonstrate this, we introduced a novel “digital twin” approach that allowed newborn chicks and ViTs to be trained in the same environments and tested with the same images. This approach made it possible—for the first time—to directly compare the learning abilities of animals and machines. We found that ViTs are not more data hungry than chicks: when ViTs are given the same training data (visual experiences) as chicks, they develop common view-invariant recognition abilities as chicks.

The Reviewers largely agreed that this is an important topic and that our “digital twin” approach has promise. However, the ratings were ambivalent for three reasons:

First, in the original submission, we only tested one ViT model (ViT-CoT). To address this critique, we added new experiments testing VideoMAE models: state-of-the-art models for temporal ViTs. As shown in Fig. 1 (rebuttal PDF), our ViT-CoT model outperformed VideoMAE by 15%. This result shows that ViT-CoT is particularly strong at learning high-level visual features in impoverished environments, akin to newborn chicks.

Second, in the original submission, we did not compare ViTs to CNNs. It was thus unclear whether the spatial inductive bias of CNNs helps or hinders performance. To address this critique, we added new experiments testing CNNs that learn via contrastive learning through time (SimCLR-CLTT; Schneider et al., 2021). As shown in Fig. 2 (rebuttal PDF), both CNNs and ViTs can learn high-level object features from impoverished visual environments. This finding shows that the inductive bias of CNNs benefits, but is not necessary for, the development of view-invariant object recognition.

Third, Reviewers questioned the scope of the project and appropriateness of the task/model system. To address these critiques, we will revise our manuscript to emphasize that our study is the first of its kind, providing a unique opportunity to directly compare the learning abilities of brains and machines. Our revision will also emphasize that newborn chicks are uniquely suited for animal-machine comparisons because chicks are the only animal that can be reared in strictly controlled environments from the onset of vision, which allowed us to fully control their training data and give that same training data to ViTs. Finally, our revision will emphasize that our results have important scientific implications. ViTs have the potential to be powerful image-computable models of newborn visual systems, but these models will not be accepted if they are thought to be more data hungry than brains. Our results directly contradict this widely held assumption, and thus, will shape our growing understanding of the relationship between brains and transformers.

---

> ### Comment · Area_Chair_ZJ86 · 2023-08-13
>
> Dear Reviewers,
>
> The authors have posted detailed individual responses to your review feedback, and the current reviewer-author discussion phase ends on Aug 21.
>
> Pls read the rebuttal(s) and respond within the **next 2-3 days**, so as to allow for further discussion (if needed).
>
> You should read the authors' rebuttal(s) in detail, and ideally:
> 1) Acknowledge the points that you find have been satisfactorily addressed;
> 2) Ask for further clarifications where needed;
> 3) Explain why you may still find certain issues to be insufficiently addressed
>
> Thanks,
> AC

---

### Decision · Program_Chairs · 2023-09-21

**Decision:**

Accept (poster)

**Comment:**

This was an unconventional submission. To compare model performance against behavioral data is not uncommon, but in this case, the data were derived from newborn chicks reared in a controlled fashion. Regardless, reviewers felt that the ideas were good, the approach was novel and interesting, and the experimental evaluation was solid.

There were initial concerns regarding novelty of the findings, the use of a supervised linear classification step and use of other self-supervised methods. The authors responded extensively, along with additional results (e.g. using VideoMAE). Reviewers generally found that their concerns were addressed and/or this AC judged the rebuttals to be convincing. For instance, the use of linear read-out classifiers is indeed a standard practice in computational neuroscience to gauge the information contained within a set of neurons.

This piece of work is best appreciated as a "scientific" one, where a specific hypothesis is tested. As with any individual scientific paper, the results form part of a larger body of work answering similar questions but under different setups or contexts. In this instance, this piece of work specifically deals with vision transformers and the controlled-rearing setup designed to test view invariance -- and this scope or these limitations should be made more explicit. But the results do add significantly to any previous findings about the data-hunger of models, and it's clearly a unique setup, which certainly adds an extra dimension.

Overall, this is an intriguing piece of work that deserves visibility. I believe the NeurIPS community will find it to be eye-opening and thought-provoking.